# Scheduling Sparse LEO Satellite Transmissions for Remote Water Level Monitoring

**DOI:** 10.3390/s23125581

**Published:** 2023-06-14

**Authors:** Garrett Kinman, Željko Žilić, David Purnell

**Affiliations:** 1Octasic Inc., 2901 Rachel, Montréal, QC H1W 4A4, Canada; garrett.kinman@mail.mcgill.ca; 2Department of Electrical and Computer Engineering, McGill University, 3480 University, Montréal, QC H3A 0E9, Canada; david.purnell.1@ulaval.ca; 3Department of Civil and Water Engineering, Laval University, pavillon Adrien-Pouliot 1065, av. de la Médecine, Québec, QC G1V 0A6, Canada

**Keywords:** Internet of Remote Things, sparse LEO satellite transmission, water-level monitoring

## Abstract

This paper explores the use of low earth orbit (LEO) satellite links in long-term monitoring of water levels across remote areas. Emerging sparse LEO satellite constellations maintain sporadic connection to the ground station, and transmissions need to be scheduled for satellite overfly periods. For remote sensing, the energy consumption optimization is critical, and we develop a learning approach for scheduling the transmission times from the sensors. Our online learning-based approach combines Monte Carlo and modified k-armed bandit approaches, to produce an inexpensive scheme that is applicable to scheduling any LEO satellite transmissions. We demonstrate its ability to adapt in three common scenarios, to save the transmission energy 20-fold, and provide the means to explore the parameters. The presented study is applicable to wide range of IoT applications in areas with no existing wireless coverages.

## 1. Introduction

The Internet of Things (IoT) has made huge advances in smart homes, industrial and other settings with numerous networking options already present. To achieve progress in the Internet of Remote Things (IoRT) for environmental monitoring in wilderness, connectivity solutions are needed that are widespread, energy-efficient and cost-efficient. This paper presents the exploration of satellite-based connectivity in the context of environmental water-level monitoring.

### 1.1. Water-Level Monitoring and Its Role in Climate

Global water-level monitoring is critical in hydrology and climate change tracking. The polar regions are arguably at the center of the climate crisis, because these regions are experiencing the most rapid changes and the largest current and future contribution to sea level rise is predicted to be from ice sheets losing mass to the ocean [1]. Predicting how the polar regions will change in the future requires field measurements, for example, from sensors that monitor changes in the atmosphere (weather stations), coastal water-level sensors, ocean buoys, or from Global Navigation Satellite System (GNSS) stations (for monitoring solid earth deformation) [1]. Despite the ever-expanding capabilities of remote sensing satellites, such measurements cannot yet be obtained from space with the same accuracy or temporal resolution as from ground-based sensors.

Climate model predictions become more reliable with an increased density of sensors, hence low-cost environmental sensor networks are emerging as a powerful tool for climate monitoring [2]. One recent innovation repurposes mass-market GNSS technology for water-level monitoring, using a technique called GNSS Interferometric Reflectometry (GNSS-IR), and has the potential to be used to increase the density of coastal water-level stations [3,4]. In remote regions such as Greenland or Antarctica, where sea level information is critical for climate monitoring [5,6], field campaigns are expensive and it may be prohibitively expensive to maintain a dense network of sensors. Wireless connectivity should reduce the maintenance cost of remote sensor networks by reducing the frequency of expensive site visits to collect data or check the status of instruments. This paper focuses on a low-cost and energy-efficient wireless communication technique using low earth orbit (LEO) satellites that is suitable for remote water-level sensor networks, notably GNSS-IR ones, Figure 1.

### 1.2. Connectivity for Internet of Remote Things

This paper addresses the problem of providing inexpensive and energy-efficient satellite IoT links in the context of GNSS-IR monitoring. Such a water-level sensor must be affordable and widely deployable. Wide geographic reach imposes the challenge of data uplink from remote locations to where that data is needed [3]. This limitation of IoT has spawned a subdomain dedicated to solving the issues of bringing IoT to the remote corners of the globe, the IoRT [7].

There is an abundance of connectivity options for IoT around populated areas. Several standard IoT connectivity options range from cellular technologies to LoRa. Connectivity options for IoRT range from low-power wide area networks (LPWANs) to low-power cellular network standards to geostationary and LEO satellites [8]. Additionally, there have been efforts into unmanned aerial vehicles supporting IoRT [8,9]. Satellite options are the only proven connectivity options for truly global coverage [8,10,11], but only if the cost and energy consumption are kept low enough. Traditional geostationary satellites are always overhead for a fixed earth location, but they are costly, require higher transmission power, and incur around 70 times longer latency than LEO satellites [7,12].

### 1.3. LEO Satellite Communications for Internet of Remote Things

For IoT applications, LEO satellites are practical for the most remote regions where terrestrial infrastructure is out of reach [7,12]. LEO communications are categorized by:Communication directness;LEO orbit configuration;By satellite service type.

Regarding communication directness, individual sensor nodes can communicate directly to a satellite (known fittingly as “direct-to-satellite”) or indirectly via a local network (often an LPWAN such as Bluetooth) centered around a satellite *gateway* [8,10]. The latter case is impractical in our case, as a gateway adds another independent hardware part not under the IoRT node control, which needs to be designed and dimensioned for multiple unknown nodes, and requires a critical mass of nodes to be useful.

Among satellite services, there are those provided by companies from the pre-IoT era (who often offer satellite internet and phone coverage as well), and the emerging sparse constellation [13] networks, such as Swarm, Lacuna, Enxaneta, Kepler or Astrocast provided by independent LEO satellite companies [8,10], which are more suited for IoT. The independent LEO satellite services use CubeSats, which are small and modular picosatellites [8]. Since IoT can tolerate intermittent connectivity better than satellite phones, their satellites can use polar orbits to provide global, but intermittent, coverage [10] by sparse constellations. In contrast, traditional LEO providers deliver continuous or near-continuous coverage using a combination of polar and non-polar orbits [10].

Using emerging *sparse LEO constellations* for IoRT has the primary benefit of requiring fewer satellites and less cost [10], but requires waiting until a satellite passes overhead [10]. To achieve low-energy LEO networking, a suitable algorithm must be devised to schedule sensing and transmissions at appropriate times such that the data is transmitted at (near-)minimal cost in energy [10].

### 1.4. Relation to Previous Work

To the best our knowledge, one previous paper [11] has examined this problem and proposed an online learning algorithm, which can learn sample-by-sample in the field, as opposed to offline in batched datasets. That work on indirect-to-satellite communications is not applicable here, as it assumes perfect knowledge of uplink availability, unpredictable multiple streams of data that can easily overflow the buffers, and is thus posed as a queue scheduling problem [11]. They then propose an online learning algorithm based on Lyupanov optimization, which is a common approach for similar queue optimization problems [11,14].

In contrast, our paper deals with direct-to-satellite communications with a known data production rate. The algorithm presented in this paper is derived from reinforcement learning, specifically Monte Carlo learning and the *k*-armed bandit problem. Because of the relative youth of the LEO satellite service (provided commercially by Swarm Technologies), an integrative approach is taken to the design from requirements, to communications technology selection, to hardware, and finally to software and algorithm design. In doing this, this paper aims to highlight key design considerations for creating a low-cost, low-power communications scheme for an IoRT device. The key contribution of this paper is the online learning-based direct-to-satellite scheduling, and associated energy model.

## 2. Materials and Methods

We implemented GNSS-IR water-level detection system on a printed circuit board (PCB), Figure 2 and deployed on Swarm network by help of our software. The PCB includes from right to left: a Swarm M138 LEO Modem, a Raspberry Pi Pico, and pads for four GNSS modules for GNSS-IR water-level measuring (together with GNSS antennas). The proposed scheduling algorithm was implemented on a dual-core ARM Cortex M processor of the Raspberry Pi Pico, where each processor core executes one process of the code. The board is sending one 128-bit message per hour, to fit within a single Swarm data plan for USD 60/year.

Figure 3 presents the schematic for the final prototype PCB design produced in this project. The left-hand side displays spaces for four GPS receivers and four GPS antenna connectors, which are the project-specific sensing components for GNSS-R water level sensing. The remaining two-thirds of the schematic are generalizeable to other projects that use the Swarm M138 LEO modem, including an mPCI-e connector, decoupling and feed-through capacitors, and headers for the microcontroller.

### 2.1. Satellite Modem Operating Specifications

For remote GNSS-IR sensors, we use the independent LEO satellite provider Swarm Technologies. The energy consumption and, consequently, transmission scheduling will depend on Swarm’s service specification and operation of the Swarm M138 modem built into our board. The Swarm modem has four operating states: (1) Sleep Mode, (2) GPS Acquisition Mode, (3) Receive Mode, and (4) Transmit Mode, Figure 4.

When the modem powers on, it enters GPS acquisition mode to determine the time and location. The modem will also re-enter GPS Acquisition Mode every 4 h or when awoken. Once a GPS fix has been acquired (30 s typical duration), the modem enters the Receive Mode, wherein it listens for a packet from any satellites passing overhead. This mode lasts until either a packet is received from a satellite (at which point it enters Transmit Mode), the modem is instructed to enter Sleep Mode, or enough time elapses that the modem automatically re-enters GPS Acquisition Mode. Robust operation and enhanced availability [15] is built into the M138 modem, as well as ensured by handling the exceptions, such as those caused by lost signals (Swarm or GNSS) or power.

If a packet is received from a satellite, the modem enters Transmit Mode, attempting to transmit queued packets and receive an acknowledgement. If successful, it will return to Receive Mode, unless put into Sleep Mode. Table 1 shows that transmission is 1–2 orders of magnitude more costly, while Sleep Mode uses 2 to 3 orders of magnitude less energy. Communication incurs a dominant part of energy consumption in IoT nodes [16]—even more so for satellite access. For instance, Swarm reports that sending a maximum-length 192-byte packet at P=2.8 W takes ΔT=3.7 s and $E_{total}=12.24$ J, while in comparison Raspberry Pico benchmark for embedded code, hello_sleep runs at 1.5 mW, as per the datasheet.

For Swarm modem’s operating modes, the energy-saving strategy includes:Keep the modem in Sleep Mode as much as possible. When not in Sleep Mode, its default state is Receive Mode, which uses much more power.Being awake dominates energy usage, either from the actual transmission energy or the GPS Acquisition and Receive Modes.Failure to transmit will waste considerable energy. Thus, one should schedule transmission to when there is a high probability of successful communication.

### 2.2. Swarm Satellite Transmission

To minimize transmission power consumption requires understanding how the satellites, transmission, and data plans work. There is not always a satellite overhead, nor are the elevation angle and environmental conditions (e.g., background RF noise) always suitable. Data rate is limited, and frequent transmissions consume energy. These factors critically impact how we orchestrate transmissions.

The nature of Swarm satellite passages is disclosed by their Web-based tool that lists upcoming satellite passes, their times, durations and max elevation angles for a given location. Elevation angles observed in Montreal, Quebec, Canada range between 15 and 85 degrees, and pass durations typically range between 10 min and an hour. In reality, even with a satellite pass, the modem might not always be able to transmit. There are many factors impacting this: satellite pass “quality”, RF background noise, environmental conditions, antenna setup, and many others.

The first factor, satellite pass quality, is due to the pass duration and maximum elevation angle. Swarm gives no guidance on what factors impair successful transmission, and one objective of this paper is for each sensor to construct an empirical model for quantifying the likelihood of a pass leading to successful transmission. The second factor, RF background noise, does have guidance provided by Swarm, Table 2, by which noise intensity of −93 dBm or lower is expected for successful transmission.

There are also the constraints imposed by Swarm data plans, priced at USD 60 per year per data plan, with up to four data plans stackable onto a single modem. Each data plan permits up to 750 packets per month, or about 25 packets per day, or about one per hour. These constraints imply that for finer temporal resolution (e.g., every 15 min), one must either bundle measurements, or pay to stack multiple data plans. The later, costly option also reduces the battery life, while bundling reduces the number of packets and possibly the cost.

The high-level view of the two main processes is shown in Figure 5. The process on the left produces and inserts the data into a circular queue. Since the Swarm modem’s internal queue can drop packets after 48 h, the circular queue needs to contain 48 h of data. Each cycle of waking from sleep, acquiring GPS, listening for a satellite, and transmitting uses a lot of extra energy. In addition, due to environmental variables, there is inherent uncertainty as to how long one can expect the modem to be awake before transmitting successfully. This precise question is examined in the rest of the paper.

### 2.3. Efficient Packet Data Bundling

Note that there are a few important functions in Figure 5, such as the data bundling, as the transmission duration directly causes energy consumption. Table 3 shows the format of data. Each datum includes a timestamp, expressed in minutes since 1 January 1970. Due to the nature of the GNSS-IR, we omit seconds, which allows the re-purposing of 4 bits for 16 status codes. For completeness, using 28 bits allows timekeeping for 510 years. This format allows the whole datum to fit in 16 bytes, which divides evenly into 192 bytes per packet, such that each packet will be maximally utilized with 12 data points per packet.

Second implicit function within the high-level processes shown in Figure 5 is that of good satellite pass selection, while the algorithm for actually predicting satellite passes—at least from the user perspective—is made fairly simple with the help of an open-source SGP4 satellite pass prediction Arduino library, quantifying what satellite passes are “good” depends much on environmental conditions, setup details, and empirical observations, as described next.

### 2.4. Online Learning Direct-to-Satellite Packet Scheduling

Transmitting to the LEO satellites can be unreliable due to minute changes in equipment setup or environmental factors. For example, severely cloudy days lead to too high RF background noise (i.e., higher than −93 dBm). Further, unshielded microcontroller within 10 to 20 cm of the antenna could increase measured RF background noise by as much as 5 to 10 dB. Further, slightly angling the antenna towards or away from a cell tower a few kilometers away could vary the RF background noise by several dB. With all these factors, creating a generalizable pass model is intractable.

Previous work with indirect-to-satellite scheduling shows that online learning is a successful strategy [11]. Thus, each individual sensor should learn for itself and for its exact site conditions and hardware setup via online learning. Previous work in indirect-to-satellite scheduling uses a Lyupanov optimization problem for network queuing to avoid making assumptions about when new data would become available [11] while the perfect knowledge of satellite overpasses is assumed. However, the data production rate is constant in our case, so rather than treating it as a network queuing problem, we ought to predict the uplink availability. Thus, a novel approach will be used.

#### 2.4.1. Algorithmic Problem Statement

For a sensor placed in a remote location, a simple and *interpretable model* is needed to be trusted to perform as expected [17]. To achieve this, a relatively simple algorithm inspired from reinforcement learning has been devised. The goal is to learn the probability of successful transmission, given three input variables: (1) the satellite pass duration (in minutes), (2) the maximum elevation angle of the satellite pass (in degrees), and (3) the RF background noise (in dBm).

Borrowing the notation from reinforcement learning, the state space *S* is the set of all possible input variable combinations, and the action space *A* is the set of all possible actions [18]. In this case, *A* consists of the actions to transmit or not to transmit for each satellite pass with pass characteristics s∈S. Let the function *v* be the mapping of *S* to a probability of successful transmission, v:S↦[0,1], and let the policy π represent the conditional probability of choosing a particular action a∈A given a state s∈S. Hence, the policy is the mechanism for choosing which satellite pass to select, given a set of passes and their characteristics.
(1)$$\pi \left(a|s\right)=P\left(A_{t}=a|S_{t}=s\right)$$

In Equation (Equation 1), $A_{t}$ represents the action at time step *t*, and $S_{t}$ represents the state at time step *t*. Regarding the probability success mapping *V*, a natural objective is thus to approximate it with collected experience: as the system runs and has successes and failures transmitting with different states s∈S, it will converge to true probabilities of successful transmission for a given state, i.e., the value function *v* [18,19].

#### 2.4.2. Modified Monte Carlo Learning

Monte Carlo learning methods approximate a value function in a simple and interpretable way by taking the value of a state to be the average return at the end of a training episode [18,19]. In the direct-to-satellite packet scheduling, the episodes are of length one, i.e., there is no sequential decision-making, simplifying the problem. If the reward is taken to be 1 for a successful transmission and 0 for an unsuccessful transmission, then the value function can be taken to be the average rate of successful transmission from a given state. If for a given state of satellite pass characteristics and RF noise, transmission is successful 50% of the time, then the value function is 0.5.

However, Monte Carlo learning requires a discrete state space, whereas the state space for this problem is continuous, so we discretize the state space. Using the Swarm pass checker, it is known that all satellite passes shown are between 15 and 90 degrees and almost all between 10 and 60 min. Additionally, while RF noise is technically continuous, the modems only report whole numbers, e.g., −95 dBm. If only integers within the range −93 dBm (the highest noise Swarm reports success transmitting with) to −106 dBm (the lowest noise measured in this project) are considered, this is naturally discretized. Table 4 shows how the state space has been chosen to be discretized. With 5 buckets for each state variable, some simple combinatorics gives 125 unique combinations, where the total set of 125 combinations represents the discretized state space.

The remaining question is that of the policy π. Clearly, once a good approximation of the true value function is made, the policy π should exploit that knowledge to select the most promising satellite passes. At the beginning, the system will not know about a good satellite pass, and it will thus have to explore with passes of different characteristics. This is an example of the *exploration–exploitation* problem in reinforcement learning [18,19]. A common approach is to explore early on and gradually exploit more with time.

#### 2.4.3. Modified *k*-Armed Bandit

Regarding the policy for packet scheduling, there is a similarity to the *k*-armed bandit problem, whereby an agent repeatedly plays the same one-step episode. In each game, the agent has a selection of options, which may give varying stochastic rewards. The goal is to learn over time which actions give the greatest expected reward [18,19]. A common approach to this problem involves softmax (Boltzmann) exploration, which derives a set of probabilities corresponding to each possible action [20]. The action with the highest expected reward has the highest probability of selection, plus all the choices are guaranteed to sum to 1 by the design.

Our problem is slightly different from the *k*-armed bandit problem in two important ways: (1) the set of actions available to the agent in each episode is different, and (2) expected reward is not only the probability of successful transmission, but its utility in the given application, most notably the timeliness. Regarding the first point, the agent is faced with a different selection of satellite passes each episode, each with their own set of pass characteristics and times at which they occur. This problem is solvable, as the modified Monte Carlo learning methods will allow keeping track of the estimated reward of each action.

#### 2.4.4. Temporal Bounds for Packet Scheduling

Addressing the reward modeling, we apply the following reasoning. A good pass in an hour is not the same as an equally good pass occurring after 24 h because: (1) data needs to be transmitted regularly, (2) the circular queue holding data has a finite capacity, and (3) the Swarm modem will drop packets from its transmission queue after a timeout. Thus, to model he preference for more prompt transmissions, a *discount factor* λ is applied to reduce the value of later passes.

Under the data plan, each modem can transmit at most one packet per hour to remain within the budget. Since there are many satellite passes to consider, the rules are needed for the interval of packet scheduling. Such rules are shown in Figure 6, in which $t_{min}$ and $t_{max}$ are the minimum and maximum amount of time (in hours) for a satellite pass, respectively, a is a vector of satellite passes between $t_{min}$ and $t_{max}$ ($a_{i}$ is the *i*-th element of a). Further, s is a vector of states (i.e., pass characteristics and RF background noise) of satellite passes of a. Let t be a vector of midpoint times of satellite passes of a.

#### 2.4.5. Algorithmic Formulation

Since we are developing a learning approach to the LEO transmission scheduling, we will rely on the activation function for classification/learning, softmax. For the set of values $x_{j},\left\{j,1,N\right\}$, it is for each value $x_{i}$ from as:(2)$$softmax\left(x_{i}\right)=\frac{exp(x_{i})}{\sum_{j}exp\left(x_{j}\right)}.$$

Since softmax adds up to 1 across all inputs, it effectively creates a probability distribution function that disproportionally favors larger values of $x_{i}$.

Let $r_{data}$ be the data point generation rate (in data points per hour), and *bundlesize* be the number of data points that comprise a full bundle. Then, the rate of full packet bundling $r_{pkt}$ is: $r_{pkt}=\frac{r_{data}}{bundlesize}$. Let softmax(z) be the vectorized softmax function where $\mathrm{softmax}{\left(z\right)}_{i}$ is the softmax of the i-th element of z, and let v(s) the vectorized value function. We express the policy π as:(3)$$\pi \left(a_{i}|s_{i}\right)=\mathrm{softmax}{\left(\lambda^{t⊖t_{min}}⊙v\left(s\right)\right)}_{i}$$
where the ⊖ and ⊙ symbols operating on vectors t and s are the element-wise subtraction and multiplication, respectively. Equation (Equation 3) expresses the probability of selecting a satellite pass $a_{i}$ from interval $\left[t_{min},t_{max}\right]$ as the softmax of the estimated transmission success probability for the pass, multiplied by a discount factor for future passes. Pass quality and promptness will be prioritized, while still giving a chance for exploration of passes currently predicted to be worse. This preference allows Monte Carlo learning to improve the value function estimates with time.

### 2.5. Uplink Transmission Energy Model

Energy consumption modeling of communication interfaces is a complex issue, and we have relied on the existing Iridium satellite communication model [21], as well as a model for long-range terrestrial network Sigfox [22], as the closest detailed model that similarly to us relies on the published energy consumption values from the datasheets. To determine the average power consumption, we introduce the unified uplink transmission energy model. Since the stochastic nature of transmission success prohibits the derivation of a deterministic model, a probabilistic model is created to give an estimate of average power consumption. There are two key causes of transmission non-determinism: (1) whether a transmission will succeed for a given pass, and (2) if it does succeed, how long the modem will be in Receive Mode before it is able to transmit.

To build the model, let $t_{SL}$ be the mean time that the modem is in Sleep Mode, $t_{GPS}$ be the mean time the modem is in GPS Acquisition Mode, and $t_{RX}$ be the mean time the modem is in Receive Mode before transmission is successful. With typical modem power consumption values $P_{SL}$, $P_{GPS}$, and $P_{RX}$, the total energy usage in these modes over a single transmit attempt cycle, $E_{attempt}$ is:(4)$$E_{attempt}=P_{SL}t_{SL}+P_{GPS}t_{GPS}+P_{RX}t_{RX}+E_{TX}N_{pkt}$$
where $N_{pkt}$ is the number of packets transmitted in a given pass. Depending on satellite pass selection and/or previous transmission attempt successes, $N_{pkt}$ may be 1 or larger. In the case of an unsuccessful attempt, $N_{pkt}$ is 0. An expression for non-zero $N_{pkt}$ is:(5)$$N_{pkt}=\frac{r_{pkt}}{p_{success}r_{attempt}}$$
where $p_{success}$ is the transmission success probability, $r_{attempt}$ is the mean transmission attempt rate, and $r_{pkt}$ is the rate at which fully bundled packets are generated. Since $r_{attempt}$ is smaller or equal to $r_{pkt}$, $N_{pkt}$ is guaranteed to be 1 or greater because successful transmission of one packet entails successful transmission of all queued packets.

In Equation (Equation 4), also note that, while $P_{SL}$, $P_{GPS}$, $t_{GPS}$, $P_{RX}$, and $E_{TX}$ (at least for full packets) are constant, $t_{SL}$ and $t_{RX}$ are variable. Here, $t_{SL}$ represents the mean time in the Sleep Mode before making a transmission attempt, approximated as: $t_{SL}=\frac{1}{r_{attempt}}$.

The value of $t_{RX}$ depends on how long the modem waits until it receives a packet and begins the transmission, or the pass is over. For a successful transmission, the quickest case is to transmit immediately after exiting GPS Acquisition Mode. The slowest success case is to transmit at the very end of the satellite pass. The worst failure case is the modem reaching the end of a given satellite pass in Receive Mode, with no transmission. In terms of $t_{RX}$, this case and successful transmission at the very end of the pass would be approximately equal. All three cases depend on the mean pass duration, denoted as $t_{pass}$.

$E_{attempt}$ can take two forms, depending on the transmission attempt success. A success is expressed in Equation (Equation 6), where $\epsilon_{pass}$ represents the proportion of a satellite pass spent in Receive Mode before receiving a packet from the satellite and is able to transmit. For pessimistic and optimistic models, $\epsilon_{pass}$ can be treated as either 1 or 0, as these serve as the upper and lower bounds of the time in Receive Mode for a given satellite pass.
(6)$$E_{success}=P_{SL}\frac{1}{r_{attempt}}+P_{GPS}t_{GPS}+\epsilon_{pass}P_{RX}t_{pass}+E_{TX}\frac{r_{pkt}}{p_{success}r_{attempt}}$$

If the attempt is a failure, the model is represented by Equation (Equation 7). Note that there is no $\epsilon_{pass}$ value and no $N_{pkt}$, as the system will wait out a full pass without transmissions.
(7)$$E_{fail}=P_{SL}\frac{1}{r_{attempt}}+P_{GPS}t_{GPS}+P_{RX}t_{pass}$$

The above two cases can be combined into a complete model: $E_{attempt}=p_{success}E_{success}+\left(1-p_{success}\right)E_{fail},$ which expands into the following expression:(8)$$\begin{array}{c} E_{attempt}=\frac{P_{SL}}{r_{attempt}}+P_{GPS}t_{GPS}+p_{success}\left(\epsilon_{pass}P_{RX}t_{pass}+\frac{E_{TX}r_{pkt}}{p_{success}r_{attempt}}\right)\\ +\left(1-p_{success}\right)P_{RX}t_{pass} \end{array}$$
where $P_{SL}$, $P_{GPS}$, and $P_{RX}$ are all constants and given by Swarm. Similarly, location fix time $t_{GPS}$ is rather constant, reported to be about 30 s by Swarm. Furthermore, note that $r_{attempt}$ depends on site conditions, project requirements, and packet scheduling. Similarly, $p_{success}$ and $t_{pass}$ depend heavily on site conditions and packet scheduling.

### 2.6. Simulation Model for Online Learning Evaluation

Setting up a number of sensors in the representative environment is expensive in time and money. Instead, the algorithm is tested with a simulated environment, similar to the methodology chosen in previous work on indirect-to-satellite scheduling [11]. Using simulations first can demonstrate the ability of the algorithm to learn underlying unknown patterns about satellite pass qualityand tune the discount factor λ parameter. To simulate the algorithm, two key components are needed: (1) virtual transmitters with an underlying probability model for which pass qualities are likely to result in transmission, and (2) randomly generated satellite pass characteristics and RF noise data. For virtual transmitters, three conceptual preference models were created, Table 5, to see how different transmitting obstacles would affect the algorithm. Note that the preferences in Table 5 refer to the conditions required for a high likelihood of success. For example, the first preference model requires high angles, long durations, and low noise for a high likelihood of success.

To create the virtual transmitter models, a function is constructed that outputs a transmission success probability by multiplying three stretched-and-shifted sigmoid (theshold activation) curves, one for each of three preference variables from Table 5. For example, the sigmoid to represent a preference for high angles would produce a value close to 1 for high angles (e.g., 70 degrees and higher) but a value close to 0 for low angles (e.g., 30 degrees and lower). The general form of the preference models is shown in Equation (Equation 9).
(9)$$P\left(success\right)=\sigma \left(k_{\theta }\left(\theta -\theta_{0}\right)\right)\times \sigma \left(k_{d}\left(d-d_{0}\right)\right)\times \sigma \left(k_{\gamma }\left(\gamma -\gamma_{0}\right)\right)$$
where σ(x) represents the sigmoid function, θ represents the max elevation angle, *d* represents the pass duration, and γ represents the RF background noise. Note that $k_{\theta }$, $k_{d}$, $k_{\gamma }$, $\theta_{0}$, $d_{0}$, and $\gamma_{0}$ represent configurable stretching and threshold shifting constants to represent the different conceptual preference models. The values of these constants used to create the three preference models by Equation (Equation 9) are shown in Table 6.

Simulated satellite passes are presented to virtual transmitters by agents imbued with a preference model and a value function approximator. The generated pass characteristics are randomly generated: each agent is exposed to random RF background noise, a vector a of satellite passes with corresponding random midpoint times t, and random pass characteristics s (except each $s_{i}\in s$ also includes the RF noise value). The randomly generated pass characteristics are drawn from a uniform distribution, and the RF background noise values are drawn from two differing distributions:Uniform across all buckets (−107 to −93 dBm).Uniform within one bucket (−107 to −105 dBm).
to express that a given sensor may experience either a full range of RF noise, or (as expected in a remote location) a narrow sub-range of RF noise.

Each agent calculates the probabilities of selecting satellite passes from the discretized states, agents’ value function approximators, and the pass midpoints. These probabilities are calculated from the policy π and satellite passes are chosen by these probabilities. Satellite pass and the RF noise characteristics are used by the agents’ preference models for transmission success probabilities. Finally, transmission successes are determined according to the agent preference model outputs, and the process repeats.

## 3. Findings

This section summarizes the findings on the suitability of the proposed on-line learning to adapt in different scenarios, on tuning of the parameters, as well as the overall energy savings and the parameters tradeoffs. For evaluation of the transmission scheduling, the goal is to (1) demonstrate the ability of the algorithm to learn patterns behind transmission success probabilities, and (2) to make apparent how the performance gets affected by the main variables, such as the overpass duration, angle and the RF noise.

Simulations were conducted for the proposed algorithm under all three preference models in Table 5. The power data published by Swarm, as per Table 1 forms the basis of the transmitter energy consumption. The effects of random noise were expressed in two ways: noise contained in a single bucket, or across all buckets from Table 4, to model the remote and populated locations, respectively. Figure 7, Figure 8, Figure 9, Figure 10, Figure 11 and Figure 12 show the transmission success and and average time to transmit for three preference models, as training epochs (i.e., the number of times that the learning steps are applied) progress, all parameterized by the discount factor λ. For their evaluation, notable is the response to overall RF noise distribution and transmission difficulty. We observe that wider RF noise distributions negatively impact the success, as well as when it is relatively hard (or easy) to transmit, there is less room for the algorithm to make huge improvements in success rate. For example, when there are few good passes, the algorithm often has to make a choice between a mediocre and a bad pass. This is seen in the lower (but still significant) improvement for the first preference model, and lower TX success rates, as there is simply no room for much improvement.

Second to note is the difference in response to discount factors λ. In the first preference model, less likely to succeed, differing discount factors made little difference to $r_{attempt}$, as the penalty for long wait for a decent pass outweigh the cost of other poorer options. For the other two preference models, however, the discount factors closer to 1 did see significantly higher average times to attempt transmission. For certain applications, an average time to transmit of 24 h may be unacceptable. If it is necessary to keep $r_{attempt}$ higher, using a lower value of $t_{max}$ or a value of λ closer to 0 would lower the average time to transmit. A lower value of $t_{max}$ in particular forces the algorithm to only consider satellite passes within a more constrained time frame.

Regarding noise distribution, Figure 7, Figure 8, Figure 9, Figure 10, Figure 11 and Figure 12 also compare two noise models, where the RF noise values were drawn uniformly from all noise buckets or from one bucket only. This second model was explored as remote sites have largely consistent RF background noise levels. The primary impact is that the algorithm consistently learns faster and converges to higher success rates when exposed to RF noise associated with remote locations. For example, the algorithm learned in 1000 epochs for preference model 2 and fully random noise what it learned in under 300 epochs for same-bucket noise.

The most permissive conceptual preference model 3, Figure 11 and Figure 12 shows that a high baseline success rate is rapidly improved through fewer epochs of training, especially as RF noise is constrained in a single bucket, as the expectation for remote, wilderness areas noise is not to occupy all buckets. Not only that the learning will be successfull, but the results are interpretable based on the understanding that the wireless channels with less contention perform better than those found in populated areas.

### Power and Energy Consumption

We now evaluate expected transmission energy savings. Energy spent for each transmission attempt $E_{attempt}$, (Equation (Equation 8)) employs parameter values listed in Table 7. They are derived from Table 1 obtained from the datasheet values. While these values may depend on exact hardware setup (e.g., voltage supply), they are all constant, as opposed to variable terms, as detailed next.

The ranges of the variables in Equation (Equation 8) are shown in Table 8. Note that “pessimistic” refers to the boundary of the interval that results in higher average consumption $E_{attempt}$, while “optimistic” refers to the boundary of the interval with lower average consumption. A value denoted as optimistic or pessimistic does not necessarily mean a value judgement for system operation. For example, a lower $r_{attempt}$ means less frequent transmission attempts, which is good for energy consumption, but causes high average time to transmit and lost data. Furthermore, note the difference between average energy consumption and the value for $E_{attempt}$ given by Equation (Equation 8); while a low $r_{attempt}$ will result in a higher $E_{attempt}$, it will result in lower average power, as shown by Equation (Equation 10) below. The term in the denominator is the average time elapsed during a complete cycle.
(10)$$P_{avg}=\frac{E_{attempt}}{\frac{1}{r_{attempt}}+t_{GPS}+p_{success}\epsilon_{pass}t_{pass}+\left(1-p_{success}\right)t_{pass}}$$

The results of these four variables on average modem power consumption are shown in Figure 13. The two dominant factors in determining average power consumption are the success rate and average time between attempts, while the smaller proportion of time $\epsilon_{pass}$ idled in Receive mode helps to keep the power low irrespective of other variables.

The impact of the success rates on power and battery requirements is shown in Table 9 for the three preference models. Note that the $p_{success}$ and $r_{attempt}$ values are taken from the simulations, and $\epsilon_{pass}$ and $t_{pass}$ are taken as 0.5 and 25 min, respectively. For each preference model, three results are shown: (1) a baseline based on taking the earliest available passes and no scheduling, (2) another baseline based on the same average $r_{attempt}$ as the simulated results but no scheduling, and (3) the test case with scheduling and simulated average $r_{attempt}$.

For simulated baseline values, the average time to attempt is represented as the first available satellite pass, expressed by
$$t_{SL,baseline}=\frac{p_{success}}{r_{pkt}}\Rightarrow r_{attempt,baseline}=\frac{r_{pkt}}{p_{success}},$$
which represents the algorithm for determining $t_{min}$, where successful transmission lead to waiting a minimum of $\frac{1}{r_{pkt}}$ hours, while unsuccessful attempts lead to no minimum wait.

The results demonstrate that online learning direct-to-satellite packet scheduling is capable of reducing average power and battery requirements around 20 times. Note that the dominant energy saving comes from reducing average attempt frequency, although the improved success rate of the scheduling algorithm introduce significant power saving. Additionally, our direct-to-satellite packet scheduling scheme enabled us to lower the attempt frequency, as it provides a built-in mechanism for selecting future passes.

#### Tradeoff between Low Power and Learning Rate

The proposed algorithm in its current state exhibits low average attempt rate values $r_{attempt}$ (still sufficiently high for the intended application). The discount factor parameter λ is investigated in Figure 14, Figure 15 and Figure 16, as it impacts the transmission attempt rate and thus the time a node takes to learn, as well as the time to transmit and the modem power.

Figure 14 and Figure 15 show that with low values of λ, future candidate passes are all discounted to such a degree that the resultant probabilities from the softmax function show little to no preference for earlier passes. Furthermore, with high values of λ (very close to 1), we observe that future candidate passes are all so little discounted that the resultant probabilities from the softmax function show little to no preference for earlier passes. It is only for λ values from around 0.9 to 0.95 (“sweet spot”) that the values of future candidate passes are discounted such that there is an apparent preference for earlier passes, such that the transmission success and time to transmit are more favorable.

Earlier passes means more frequent transmission attempts. A downside to this is in smaller expected power savings, as more frequent transmissions require significantly more average power. There is thus a tradeoff between having a high learning rate and achieving low average power consumption, Figure 14, Figure 15 and Figure 16, with the “sweet spot” area of λ selected for all earlier experiments, e.g., Table 9.

## 4. Conclusions and Future Work

We have presented and evaluated the transmission scheduling for emerging sparse LEO satellite services suitable for IoT. This is the first published study that addresses the critical uplink availability issue with sparse LEO constellation. A detailed probabilistic energy consumption model was developed used to evaluate our on-line learning scheme for predicting transmission periods. Our learning proposal is inexpensive computationally, learns in small increments and in a modest number of training epochs, and is interpretable, unlike most modern machine learning approaches. For three common scenarios, we have observed up to 20-fold reduction in power and battery requirements of the transmissions attributable to this learning approach. Since the whole scheme is suitable for the smallest embedded microcontrollers, we have demonstrated its implementation on Raspberry Pi Pico that manages to pack all sensed data within the least expensive dataplan.

While the intended application is the water-level monitoring in remote locations, the proposed scheme is practical for other IoRT applications, as it incurs modest computing costs. The results demonstrate the ability of the proposal to facilitate energy-efficient data collection over prolonged periods of time.

The whole analysis was done by simulations to reduce the burden in cost, time and also to combat location-dependence, since RF noise in the city differs from the intended remote locations; while we demonstrate the ability of our proposed scheme to adapt appropriately, obtain interpretable learning and explore the impact of variables on the performance of the algorithm, they are nonetheless simulated results.

Complete code and data for modeling, learning and energy consumption evaluation is made available by GitHub (as reported below in “Data Availability” statement). The code and data can be readily used for further exploration of sparse LEO connectivity for IoT in general.

### 4.1. Future Work

Related to the learning approach, further research could examine the possibility of adaptive learning rates. One possibility for polar regions would be increasing the transmission attempt frequency (e.g., by decreasing the discount factor λ) during the summer when solar power is abundant, and lowering the transmission attempt frequency during the polar winters.

There are two natural steps that could be taken to improve the simulation setup in future: (1) the simulations could be based on extracted real-world data for a given application, or (2) multiple sensors could be placed in the field for weeks or months. The first option is natural, as the risk of needing to adjust the algorithm is high, making tuning in simulations and testing on hardware only when confident in good results is the best route.

#### 4.1.1. Potential Simulation Model Improvements

The most apparent way to improve the simulations is to generate a more representative RF noise distribution and to use the Swarm satellite pass information, something which is not available in the remote locations. For the latter, there are the versions of the online satellite pass library, as used in the open source SGP4 Arduino library or in Python [23,24].

#### 4.1.2. RF Noise Impact on Prototype System

One factor discovered in verification testing of a prototype is the sensitivity of the antenna to noise. For example, minor changes in the exact positioning of the PCB and microcontroller under the ground plane could vary the measured noise by as much as 3 dB. Clearly, the antenna receives RF interference from an unshielded device in the immediate vicinity (e.g., 10 to 20 cm). In fact, it was precisely this interference that played a role in the design of a online learning direct-to-satellite packet scheduling algorithm; since no method could possibly account for the range of possible housing, ground plane, and even power supply designs, as well as varied site conditions and lines of sight, there could be no single ultimate “good” pass model, so a learning-based approach is used.

## Figures and Tables

**Figure 1 sensors-23-05581-f001:**
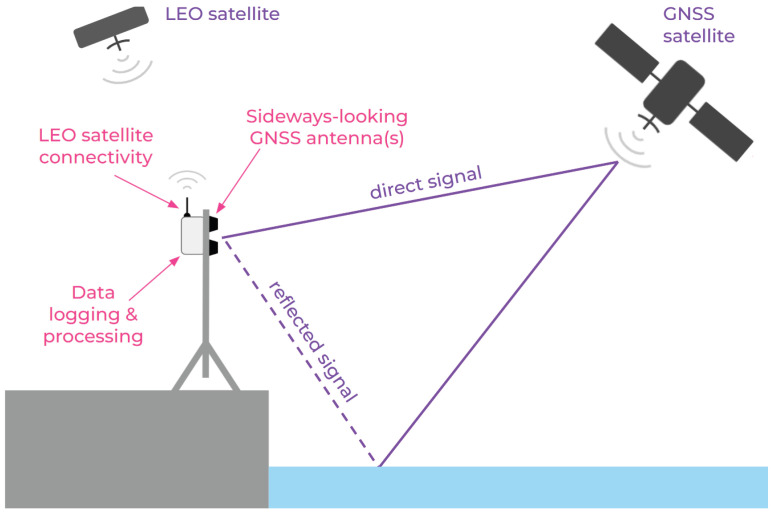
Schematic showing a GNSS-IR water-level sensor with integrated LEO satellite connectivity.

**Figure 2 sensors-23-05581-f002:**
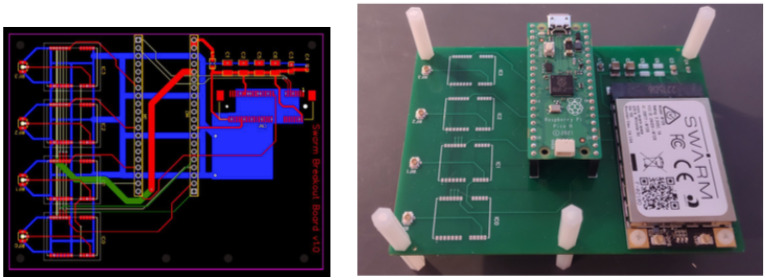
Board design: PCB layers and board populated with Raspberry and Swarm modules.

**Figure 3 sensors-23-05581-f003:**
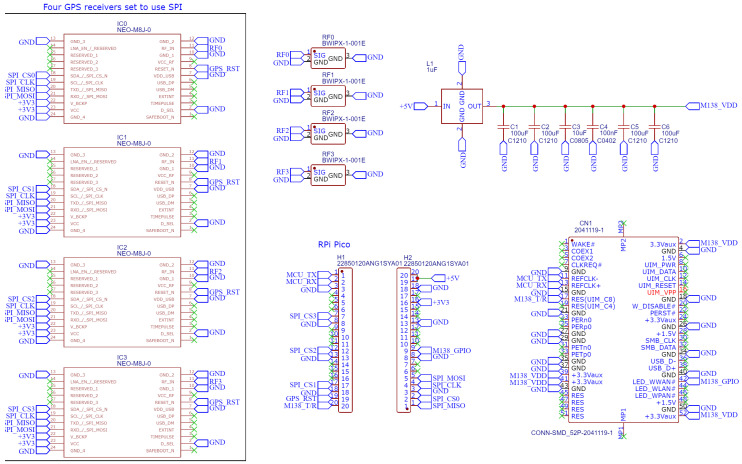
Board schematic with GNSS receivers, Raspberry Pico microcontroller and Swarm modem.

**Figure 4 sensors-23-05581-f004:**
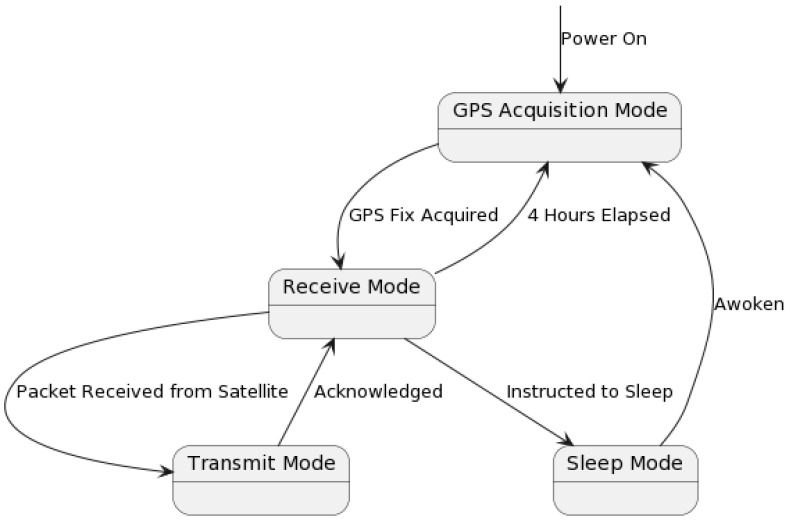
State machine of the Swarm M138 operating modes.

**Figure 5 sensors-23-05581-f005:**
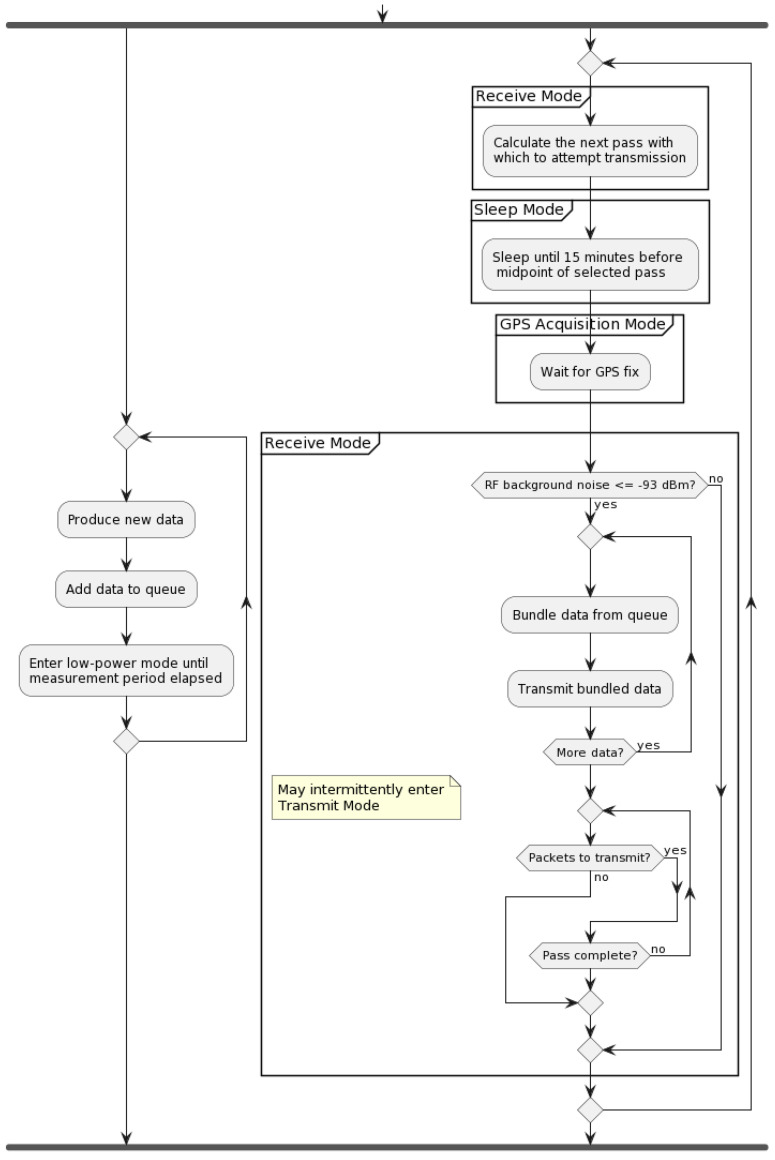
Activity diagram for the host device with two processes.

**Figure 6 sensors-23-05581-f006:**
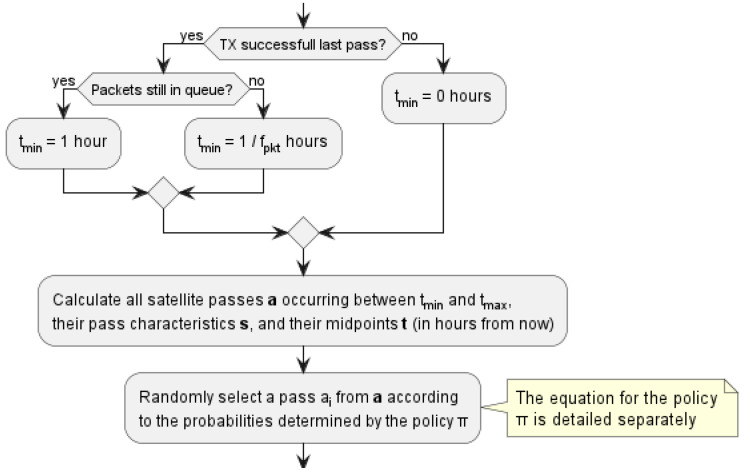
Routine for selecting the next satellite pass to attempt transmission.

**Figure 7 sensors-23-05581-f007:**
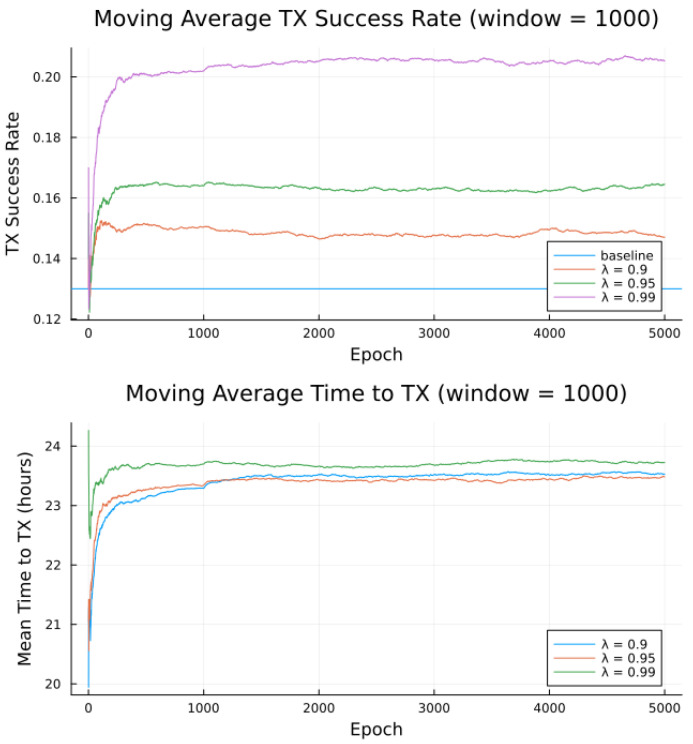
Simulated results for **preference model 1** and random noise within **1 bucket**.

**Figure 8 sensors-23-05581-f008:**
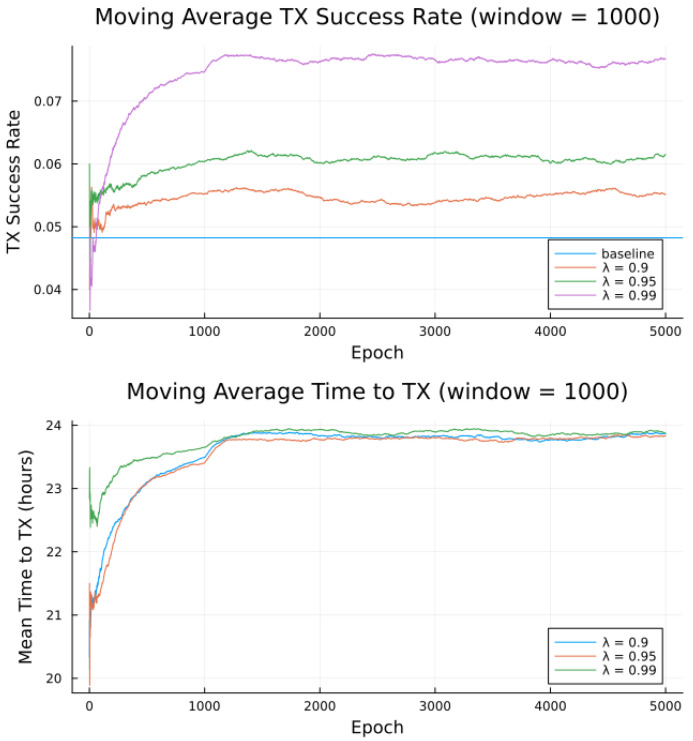
Simulated results for **preference model 1** and random noise across **all buckets**.

**Figure 9 sensors-23-05581-f009:**
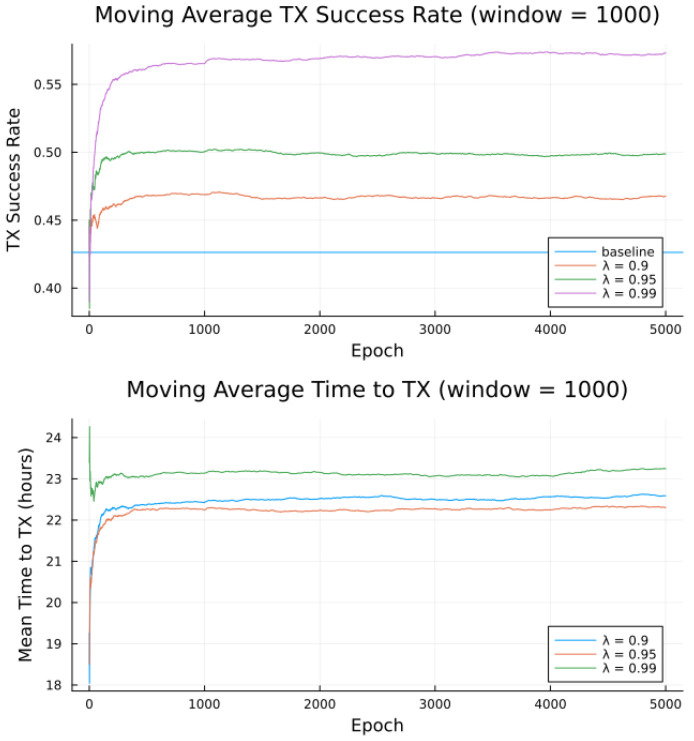
Simulated results for **preference model 2** and random noise within **1 bucket**.

**Figure 10 sensors-23-05581-f010:**
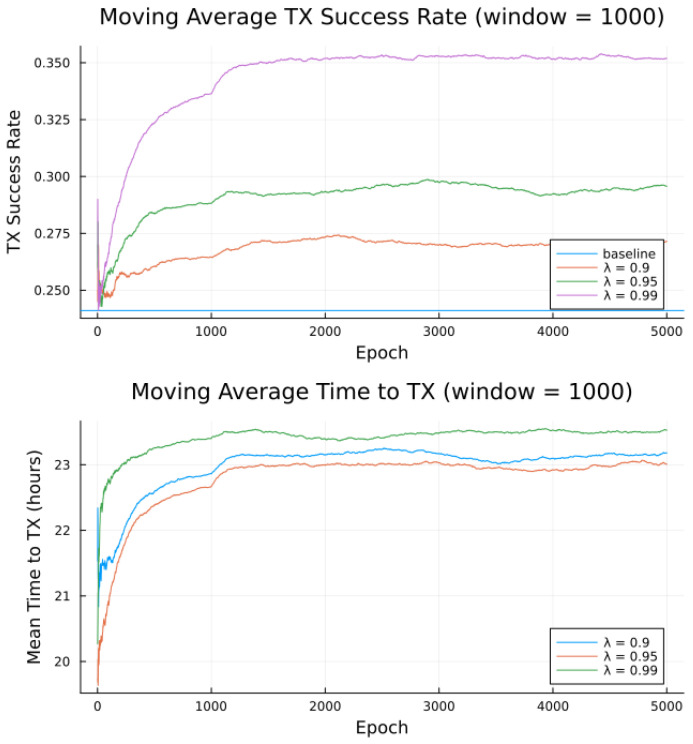
Simulated results for **preference model 2** and random noise across **all buckets**.

**Figure 11 sensors-23-05581-f011:**
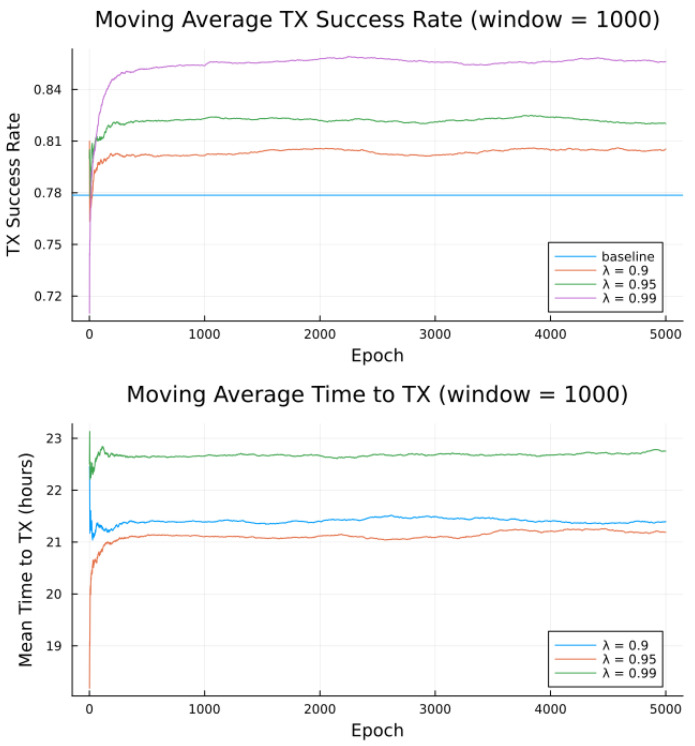
Simulated results for **preference model 3** and random noise within **1 bucket**.

**Figure 12 sensors-23-05581-f012:**
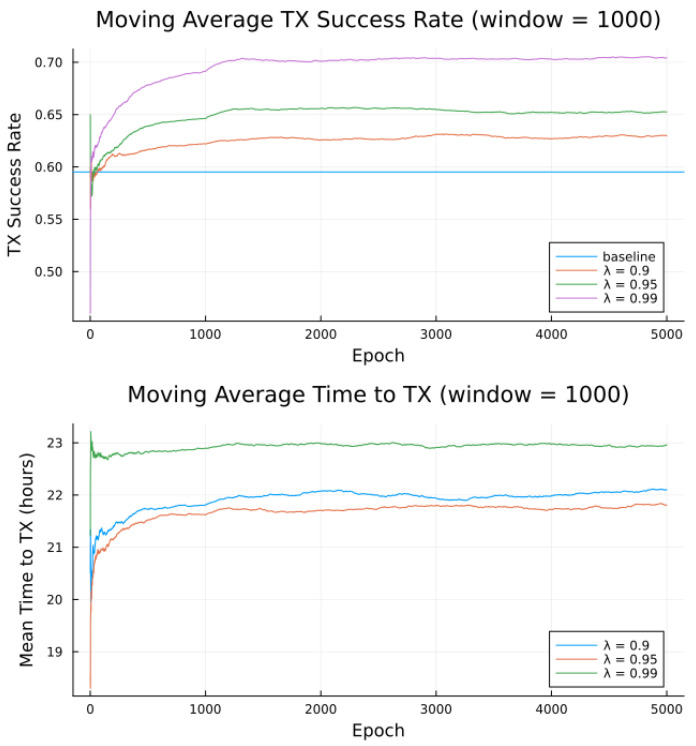
Simulated results for **preference model 3** and random noise across **all buckets**.

**Figure 13 sensors-23-05581-f013:**
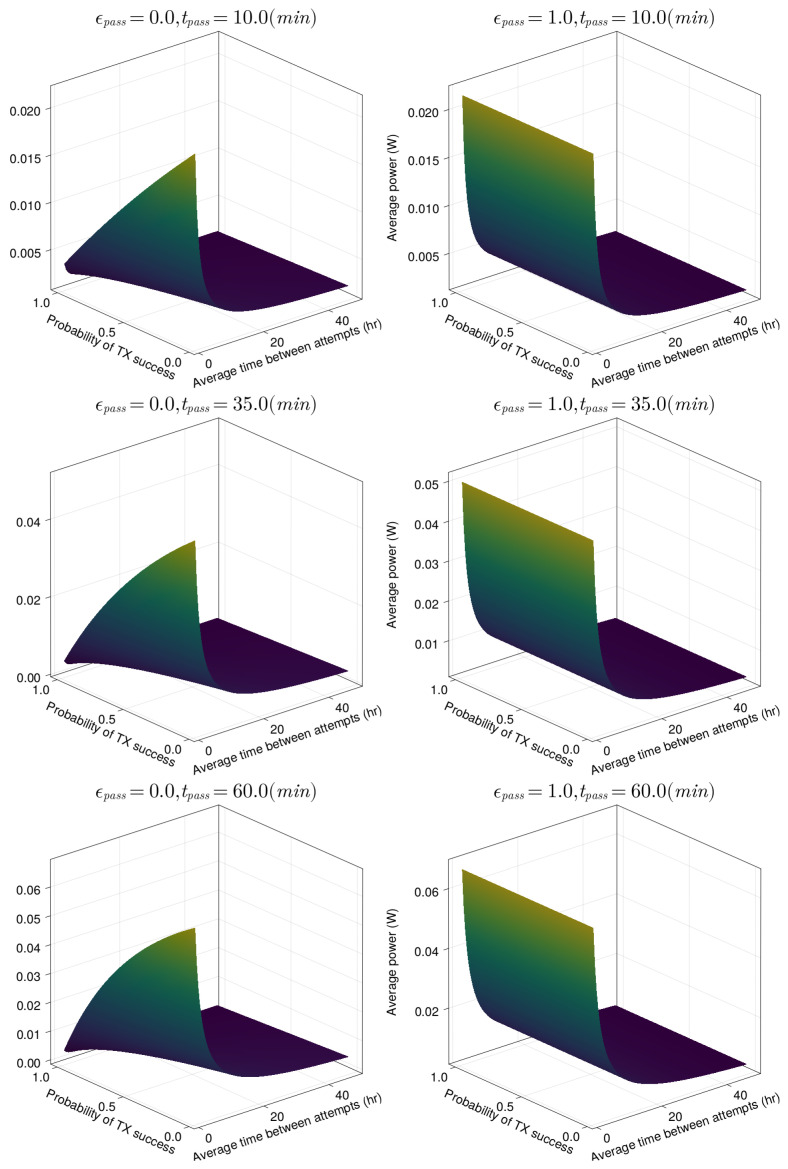
Average power of the Swarm modem under different pass variable values.

**Figure 14 sensors-23-05581-f014:**
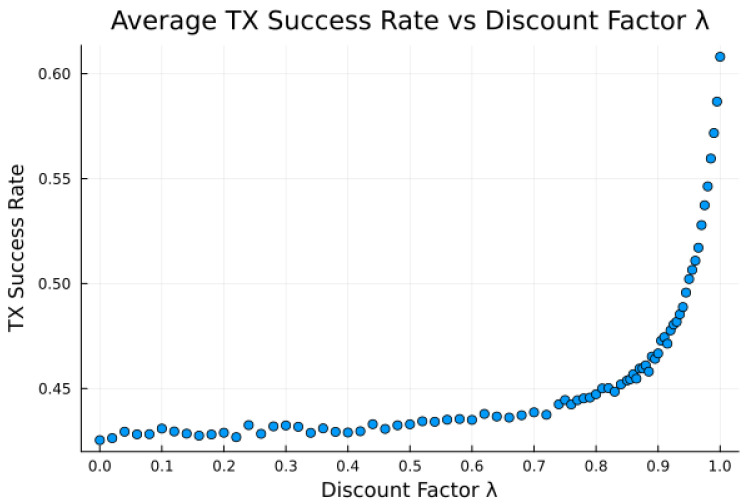
Average success rate versus discount factor λ for bucketed noise and moderate preference model 2.

**Figure 15 sensors-23-05581-f015:**
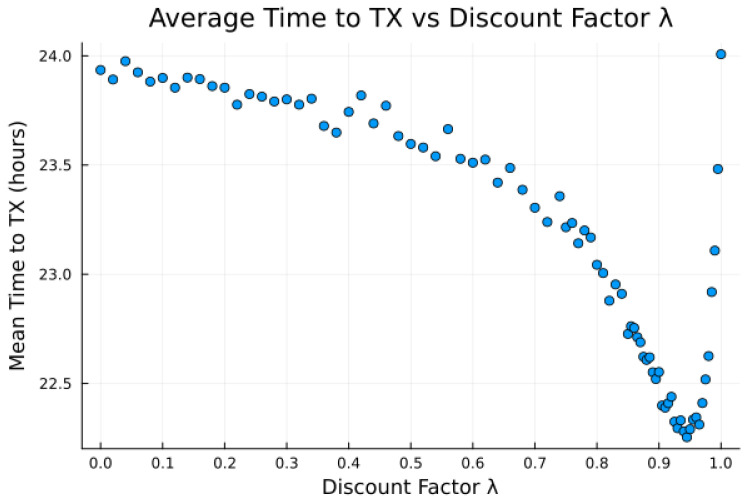
Average time to attempt transmission versus discount factor λ for bucketed noise and moderate preference model 2.

**Figure 16 sensors-23-05581-f016:**
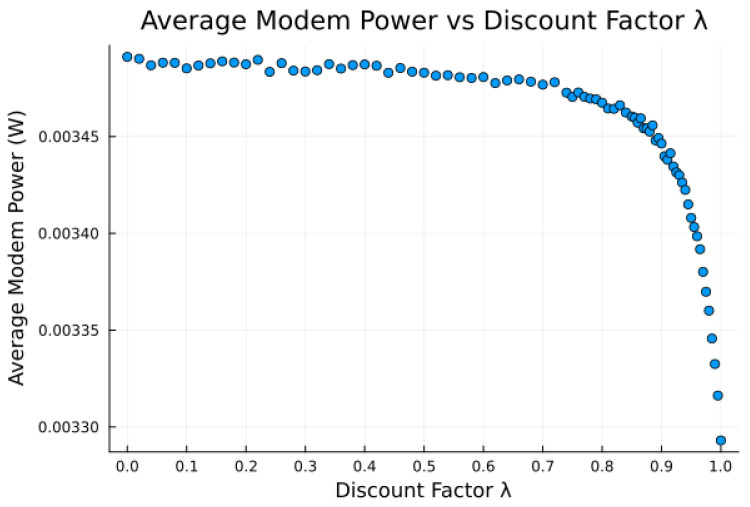
Average modem power versus discount factor λ for bucketed noise and moderate preference model 2, using $\epsilon_{pass}=0.5$, $r_{attempt}=\frac{1}{24}\;h^{-1}$, and $t_{pass}=25\;\min$.

**Table 1 sensors-23-05581-t001:** DC power characteristics of 4 modes of operation for 5V/3.3V power supply.

Mode	Typical Current at 5 V/3.3 V	Typical Power at 5 V/3.3 V
Transmit	550 mA/850 mA	2.8 W/2.8 W
GPS Acquisition	45 mA/45 mA	230 mW/150 mW
Receive	26 mA/26 mA	130 mW/86 mW
Sleep	<110 μA/80 μA	<550 μW/260 μW

**Table 2 sensors-23-05581-t002:** Background noise intensity required for likelihood of transmission.

Background Noise RSSI (dBm)	Quality (for Transmission)
−90 and higher	Bad (unlikely to work)
−93 and lower	Marginal
−97 and lower	OK
−100 and lower	Good
−105 and lower	Great

**Table 3 sensors-23-05581-t003:** Format for each datum within the software.

Name	Type	Bits
Water level	Floating-point	32 bits (4 bytes)
Error	Floating-point	32 bits (4 bytes)
Roughness	Floating-point	32 bits (4 bytes)
Minutes since 1 January 1970	Positive integer	28 bits (<4 bytes)
Status	Positive integer	4 bits (<1 byte)
Total		128 bits (16 bytes)

**Table 4 sensors-23-05581-t004:** State space bucketing for each state variable.

Bucket Number	Max Elevation Angle (°)	Pass Duration (minutes)	RF Background Noise (dBm)
1	15 to 30	10 to 20	−93 to −95
2	31 to 45	21 to 30	−96 to −98
3	46 to 60	31 to 40	−99 to −101
4	61 to 75	41 to 50	−102 to −104
5	76 to 90	51 and higher	−105 and lower

**Table 5 sensors-23-05581-t005:** Conceptual preference models for the virtual transmitters.

Model	Elevation Angle	Pass Duration	RF Background Noise
1	High angles	Long time	Low noise
2	Mid to high angles	Mid to long time	Low to mid noise
3	Low to high angles	Short to long time	Low to high noise

**Table 6 sensors-23-05581-t006:** Constants for the three preference models.

Preference Model	$k_{\theta }$	$\theta_{0}$	$k_{d}$	$d_{0}$	$k_{\gamma }$	$\gamma_{0}$
1	0.5	70	0.5	35	−1	−102
2	0.5	50	0.5	20	−1	−99
3	0.5	30	0.5	10	−1	−96

**Table 7 sensors-23-05581-t007:** Constants used for transmission attempt energy model.

Constant	Meaning	Value
$P_{SL}$	Sleep Power	550 μW
$P_{GPS}$	GPS Power	230 mW
$t_{GPS}$	Time to GPS Fix	30 s
$P_{RX}$	Receive Power	130 mW
$E_{TX}$	Transmit Energy	12.24 J
$r_{pkt}$	Packet Rate	$\frac{1}{3}\;h^{-1}$

**Table 8 sensors-23-05581-t008:** Sample variable ranges for transmission attempt energy model.

Parameter	Variable	Pessimistic Value	Optimistic Value
Attempt rate	$r_{attempt}$	$1\;h^{-1}$	$\frac{1}{48}\;h^{-1}$
Success probability	$p_{success}$	0.0	1.0
Portion of time in read mode	$\epsilon_{pass}$	1.0	0.0
Overpass duration	$t_{pass}$	60 min	10 min

**Table 9 sensors-23-05581-t009:** Sample energy savings from simulated packet scheduling for a year of operation.

Preference Model	Success Prob. $p_{\mathrm{success}}$	Attempt Rate $r_{\mathrm{attempt}}$	Average Power	Required Battery Capacity
1	0.13	$2.564\;h^{-1}$	67.54 mW	592.1 Wh
	0.13	$\frac{1}{24}\;h^{-1}$	3.810 mW	33.40 Wh
	0.20	$\frac{1}{24}\;h^{-1}$	3.735 mW	32.74 Wh
2	0.42	$0.7937\;h^{-1}$	29.31 mW	256.9 Wh
	0.42	$\frac{1}{23}\;h^{-1}$	3.575 mW	31.34 Wh
	0.57	$\frac{1}{23}\;h^{-1}$	3.405 mW	29.85 Wh
3	0.78	$0.4274\;h^{-1}$	14.95 mW	131.1 Wh
	0.78	$\frac{1}{22}\;h^{-1}$	3.234 mW	28.35 Wh
	0.85	$\frac{1}{22}\;h^{-1}$	3.151 mW	27.62 Wh

## Data Availability

The data used in this study was generated by a simulation program placed in the repository: https://github.com/garrettkinman/Self-Learning-Satellite-Pass-Selection (accessed on 31 May 2023).

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
