# Peer review of "Scheduling Sparse LEO Satellite Transmissions for Remote Water Level Monitoring"

_sensors, 2023, doi:10.3390/s23125581_

Round 1

Reviewer 1 Report

The paper focuses on the problem of providing inexpensive and energy-efficient satellite IoT links for water monitoring in remote areas, assisted by LEO satellite communications. In this context, water-level stations can be more remotely and widely deployed, leading to more accurate sensing and thus enhanced water prediction models.

The paper is generally well-written, and the concepts and methods are well-explained and factually correct. It is also very interesting and addresses a very relevant problem. However, I have some concerns that need to be addressed before I can recommend the manuscript for publication.

1. The introduction fails to accurately state the problem of interest and how it will be addressed. L73-75 state: "but requires data buffering until the satellite passes overhead 73 [9]. To achieve low-energy LEO networking,..." My concern is that if the goal is to optimize energy transmission efficiency in the uplink, regardless of energy costs for buffering operations, buffering is also an alternative to optimize transmission efficiency in the case of discontinuous satellite passes. So, this raises the question: Is the proposed online-learning scheduling algorithm a better alternative? To me, this is not self-evident. Note that online learning also comes at an energy price (which the authors do not take into account). Putting aside that, buffering is much more stable and accurate than online learning-based predictions. So, judging energy efficiency with a simple energy model won't be enough to prove the reliability of the proposed algorithm.

Since this is a complex problem, I suggest the authors do their best to explain the general picture and better motivate their approach, highlighting the features of the approach and advantages compared to buffering-based algorithms for scheduling.

2. Following the previous point, to better state the problem and how it will be solved, it must be self-evident how the energy efficiency will be measured and how the online scheduling algorithm will optimize such a metric.

3. Similarly, L78-86 explains similar works and how this study is different. However, I suggest the authors better motivate why they consider it important to study direct-to-satellite communication as opposed to indirect-to-satellite communications. It seems to me that the latter can be more reliable and not necessarily less energy efficient.

4. Since the energy model in Sec. 2.5 plays an important role in measuring efficiency, the authors should better justify their modeling assumptions, grounding them in the literature. I believe accurately measuring transmission energy efficiency is itself a fundamental problem.

5. Please improve the experimental part by including a plot illustrating the tradeoff between low power and learning rate (Sec. 4.1.1).

6. In conclusions and future work, include a discussion of the online learning and inference algorithmic complexity and energy costs.

7. Clearly state in the abstract the methods, type of experiments with simulated data, and main conclusions.

Author Response

The response document is attached.

Reviewer 2 Report

Abstract

The abstract does not provide any information on the results or the effectiveness of the learning approach developed for scheduling transmission times from the sensors. It also does not mention the specific water, ice, and snow levels that are being monitored or the geographic areas where the monitoring is taking place. Providing this information would help readers better understand the context and significance of the study.

Introduction

The introduction does not clearly state the research question or objective of the study. It briefly mentions the need for satellite-based connectivity for environmental monitoring in remote areas, specifically for water-level monitoring, but it does not explicitly state the research question or the problem that the study aims to address. The introduction could benefit from a clear statement of the research question or objective to guide the reader and provide context for the study's significance.

Concerning previous studies, the authors reported on previous studies related to the use of satellite-based connectivity for environmental monitoring in remote areas, as well as the challenges and benefits of using LEO satellites for IoT applications. The authors also discussed a previous study that proposed an online learning algorithm for scheduling transmissions in indirect-to-satellite communications. The authors compared their study to this previous work and highlighted the differences in their approach and contributions. Overall, the authors provided a thorough review of relevant literature to contextualize their study.

Methodology

The methodology is missing a few key elements:

1.      Details on the hardware and sensors used. No specifics are given on the modems, antennas, IR sensors, microcontrollers, etc.

2.      Experimental validation. No details are provided on how the algorithm was tested and evaluated. Were actual sensors deployed in the field? Or was the algorithm only simulated?

3.      Parameters tuning. There is little information on how the parameters like the discount factor (lambda) and bucket sizes were chosen. Some hyperparameter optimization and sensitivity analysis would strengthen the methodology.

4.      Failure modes and recovery. There is no discussion of what happens when transmissions fail, packets are dropped, modems lose GPS fix, etc. More robustness is needed for a practical methodology.

5.      Data plan optimization. While data plans are mentioned, there is no discussion of optimizing data usage to minimize costs and extend battery life. This could be a key part of the methodology.

In summary, the paper focuses more on the theoretical algorithm rather than a full methodology for deploying and operating the system in practice. More details and validation around the practical aspects would strengthen the materials and methods section.

Findings

The study findings are missing a section discussing the results and key takeaways. Some important points that seem to be missing:

·         A summary of the main trends and observations from the simulation results

·         Key insights into how the algorithm performance was impacted by factors like overpass duration, angle, and RF noise

·         High-level conclusions about the effectiveness of the scheduling algorithm in improving transmission success rate and reducing energy consumption

Without these result discussions, the presented simulation figures and energy model calculations lack context and interpretation. The reader is left to derive their own inferences from the raw data, whereas a proper results section would tie the findings together and highlight the most important implications.

The current sections focus heavily on describing the experimental setup, variables, and equations used. But to truly evaluate the proposed transmission scheduling algorithm, there needs to be a thorough analysis of what the simulation outputs actually mean in practice.

In summary, to improve the study findings, the authors could benefit from adding a results section that discusses the following:

1.      Overall trends in the simulation results

2.      Key impacts of different variables on algorithm performance

3.      High-level conclusions about the scheduling algorithm's effectiveness

4.      Practical implications and takeaways based on the presented data

Moreover, concerning comparison of the study’s findings with that of previous studies, based on the information provided, it does not appear that the authors compared their findings with those of previous studies. There is no mention of previous work, related research, or how the current study builds upon the state of the art.

The authors should know that, comparing results with prior studies is an important part of research. It helps establish:

1.      Whether the current findings support or contradict what has been found before

2.      The relative performance and effectiveness of the proposed approach compared to existing techniques

3.      Areas where the current study makes improvements over previous work

4.      Limitations and open problems that still need to be addressed

Without such comparisons, it is difficult to gauge how the proposed scheduling algorithm and the obtained results fit within the broader research landscape. It is unclear whether the findings represent a significant step forward or are largely in line with what has already been achieved.

Comparing to related work would typically be done in one of two sections:

1.      Related Work - To provide context and position the study within the current state of research

2.      Results and Discussion - To interpret the obtained results in light of what has been found before

So in short, based on the information provided, it seems the authors have not yet compared their findings and results to those of previous studies in the field. Such comparisons would help establish the significance and novelty of the work, as well as its limitations.

Conclusion and future research direction

The conclusions and future work section is missing several key elements:

1.      A clear summary of the main conclusions from the study. Given the length of the paper, a 1-2 paragraph summary highlighting the most important takeaways would be useful.

2.      A discussion of the limitations of the current work. All research has limitations, and acknowledging them helps frame the implications and significance of the findings.

3.      Suggestions for applications and use cases where the proposed approach could be helpful. Knowing how the scheduling algorithm could be deployed in practice makes the research more impactful.

4.      Specific directions for future research beyond "improve simulations" and "reduce power tradeoff." The authors could propose:

·         Extending the model to include multiple satellites and satellite constellations

·         Optimizing the scheduling for different types of sensor networks and IoT applications

·         Investigating alternative learning and scheduling techniques for comparison

·         Studying performance under more realistic RF noise and satellite pass conditions

5.      A final concluding statement that summarizes the value and importance of the proposed scheduling algorithm, given the current state of the art.

In summary, the main components missing are:

·         A concise summary of key conclusions

·         A discussion of limitations

·         Suggestions for useful applications

·         Specific proposals for future research beyond generic "improvements"

·         A final concluding statement

These elements would help reinforce the importance and implications of the study. The current section focuses more on future simulation improvements and potential tradeoffs rather than providing an overall perspective on the value and limitations of the research.

Review comment summary

Scheduling LEO Satellite Transmissions for Remote Water Level Monitoring

The proposed transmission scheduling algorithm shows promise, but the paper has gaps that limit its impact:

·         Results are presented with little interpretation and conclusions. Understanding trends, variable impacts, and findings are missing.

·         No comparison is made to prior work. Without establishing novelty, significance is diminished.

·         Conclusions lack a summary, discussion of limitations, suggestions for applications, and specific future work proposals.

·         Methods are described in detail, but results are under-discussed. A balanced treatment is needed.

·         Practical value, feasibility, and limitations of deploying the approach are hardly discussed.

Major revisions are recommended:

·         Thoroughly interpret and discuss results

·         Compare findings to relevant prior work

·         Balance methods and results sections

·         Discuss real-world value, feasibility, and limitations

·         Strengthen conclusions

 Minor editing of the English language required

Author Response

The response document is attached.

Round 2

Reviewer 1 Report

Thank you for taking the time to address my questions and concerns. The subject and work are very interesting and timely. I think the manuscript now reads very well and the paper will turn out to be a very valuable contribution. 

Author Response

Thank you. 

Reviewer 2 Report

The authors have addressed all the comments made in the first round of the review. 

Author Response

Thank you.